# Upgrading of Biobased Glycerol to Glycerol Carbonate as a Tool to Reduce the CO_2_ Emissions of the Biodiesel Fuel Life Cycle

**DOI:** 10.3390/bioengineering9120778

**Published:** 2022-12-06

**Authors:** Biagio Anderlini, Alberto Ughetti, Emma Cristoni, Luca Forti, Luca Rigamonti, Fabrizio Roncaglia

**Affiliations:** 1Department of Chemical and Geological Sciences, University of Modena and Reggio Emilia, Via G. Campi 103, 41125 Modena, Italy; 2Department of Life Sciences, University of Modena and Reggio Emilia, Via G. Campi 103, 41125 Modena, Italy; 3Interdepartmental Centre H2-MORE, University of Modena and Reggio Emilia, Via Università 4, 41121 Modena, Italy; 4INSTM Research Unit of Modena, Via G. Campi 103, 41125 Modena, Italy

**Keywords:** biodiesel, glycerol, flow chemistry, glycerol carbonate, carbon cycle, CO_2_ capture

## Abstract

With regards to oil-based diesel fuel, the adoption of bio-derived diesel fuel was estimated to reduce CO_2_ emissions by approximately 75%, considering the whole life cycle. In this paper, we present a novel continuous-flow process able to transfer an equimolar amount of CO_2_ (through urea) to glycerol, producing glycerol carbonate. This represents a convenient tool, able to both improve the efficiency of the biodiesel production through the conversion of waste streams into added-value chemicals and to beneficially contribute to the whole carbon cycle. By means of a Design of Experiments approach, the influence of key operating variables on the product yield was studied and statistically modeled.

## 1. Introduction

In addition to different natural dynamics outside of our control, such as solar activity, developments over the last hundred years have been accompanied by a general and progressive increase in the mean global temperature, collectively known as ‘global warming’. This variation is strictly related to several climate alterations such as melting glaciers and rising sea levels, but also to extreme weather events, a change in wildlife habitats, and an array of other impacts [1]. Beyond the heating due to direct irradiation from the sun, the atmosphere plays a determinant role in retaining part of the thermal energy close to the Earth’s surface [1]. This heat-trapping phenomenon, known as the “greenhouse effect”, comes from some naturally occurring gases, such as CO_2_, methane, and nitrous oxides, and helps our planet to maintain stable conditions for life.

The continuous and ever-increasing exploitation of fossil carbon resources to sustain our development brought a symmetrical increase in greenhouse gas (GHG) emissions, especially CO_2_, which have been linked to an increased greenhouse effect.

More sustainable strategies to drive our development without hampering the progress of future generations are urgently required and, in this context, a great deal of attention has recently been devoted to the study and implementation of biorefineries. Biorefineries are production facilities able to exploit renewable biomass instead of a fossil carbon resource and convert it into carbon-based products, such materials, fuels, or energy [2]. CO_2_ fixation during plant growth can considerably reduce the impact on carbon balance when a vegetable source is exploited. This provides biorefineries with a greatly improved GHG emissions profile compared to traditional refineries. Among the different biorefinery types, biodiesel biorefineries focus on the production of carboxylic acid esters that can be directly blended with regular diesel fuel, starting from vegetable or animal fats. The latter are subjected to acid- or base-catalyzed transesterification in the presence of an excess of a short chain alcohol, to give a mixture of fatty acid methyl (or ethyl) esters, which is biodiesel. Depending on the triglyceride source, it has been estimated that the adoption of bio-derived diesel fuel brings between a 63% (energy crops) and 86% (waste fat) reduction in GHG emission compared to oil-based diesel fuel [3]. Moreover, a common factor in all transesterification processes is the co-production of glycerol (Gly) that is usually considered a waste stream, especially in small-scale productions where further refining costs are not justified. The present crude Gly price is as low as between USD 0.04 and 0.09/lb and it is expected to decrease further according to the growing industrial availability. Current global production is approximately 42 billion liters per year [4].

Gly shows some direct potential uses in the pharma, food, cosmetic, and polymer industries; however, its features (limited miscibility with organics, high boiling point, chemoselectivity of primary vs. secondary hydroxyl), as well as the presence of contaminants in the raw material, pose several barriers to its free applicability. As a result, the identification of effective processes able to economically convert Gly into value-added chemicals represents a key development needed to improve the sustainability of the whole biodiesel industry [5].

Various Gly-derived upgraded structures with enhanced properties and extended application profiles are well-described. Solketal [6] and other acetals, 1,3-propanediol [7,8] and dihydroxy acetone [9], represent known examples, and among these, a leading position is occupied by glycerol carbonate (GC), which represents the main focus of this work. As a multiple-site electrophile, GC can efficiently and selectively interact with diverse nucleophiles, such as amines [10,11,12], amino acids [13,14], phenols [15], carboxylates [16], and others [17]. The nucleophilic property of the free hydroxyl function can be directly exploited [18,19] or can be used to obtain a new electrophilic site capable of additional reactivity [20,21]. Valuable GC-derived oligomers or polymers, especially polyglycerols [22,23], polycarbonates, and polyesters [24], find application in biomedical tissue engineering [25] and in the controlled release of active pharmaceutical ingredients [26]. Polyglycerols are also applied in water-dispersible polymers, adhesives [27,28], and bio-based surfactants [16]. As a building block, GC has been included in diverse classes of polymeric materials, such as acrylates [29,30], polyesters [31,32,33], isocyanate-free polyurethanes [33,34,35,36,37], hybrid structures [38,39], and others [40,41]. Finally, GC can also be used as a carbonate carrier able to transfer the same function to other polyols, including sugars [42,43], through transcarbonylation processes. The carbonate function can also be considered as a precursor to highly electrophilic (and valuable) epoxy function [44,45].

Thanks to the excellent properties of GC, a number of methods able to convert Gly into GC were devised [46,47]. Based on the nature of the main interacting reagent involved, they can be divided into three groups, as follows:
(i).processes using activated phosgene-sourced reagents, such as phosgene itself, chloroformates, or carbonyldiimidazole;(ii).processes using activated reagents not sourced from phosgene, such as dialkyl carbonates, diaryl carbonates, or CO + O_2_;(iii).processes using non-activated CO_2_-sourced reagents, such as CO_2_ itself or urea.

Class (i) processes are characterized by high Gly conversion and product selectivity, but also by health and environmental issues due to the toxicity of the involved reagent (or precursor). The reactivity of class (ii) reagents allows high efficiency as well and, thanks to the better sustainability profile, the related processes are the most investigated [48,49,50]. Different implementations of these processes are described, extending through various catalysts and plant engineering, including continuous-flow techniques [51,52]. Organic carbonates can, in principle, be prepared from CO_2_, however, aside from promising research advancements [53,54], no industrial process is currently available, possibly due to known issues such as process reversibility and hydrolytic instability of the product [55].

Dealing with class (iii) reagents, CO_2_ certainly represents the more atom economical choice, but the same hurdles described to form alkyl carbonates also hamper the direct carbonylation of glycerol to GC [56,57]. The option to use highly toxic glycidol as an activated Gly-sourced substrate [58,59] able to overcome the chemical inertness of CO_2_ presents critical toxicity features similar to phosgene or gaseous CO [60]. To pursue chemical upgrading routes characterized by a non-toxic and stable reactant as well as featuring high atom economy, our attention was devoted to urea. Urea is a white crystalline solid containing 46 wt% of nitrogen and, being non-toxic, is largely used as an animal feed additive and fertilizer [61]. It is even more resonance-stabilized [62] than CO_2_, thus, its reactivity with polyols is expected to not be very manifest; in fact, thermally stable mixtures between urea and polyols are well-described [63] and are used as deep eutectic solvents. What makes urea an attractive carbonylation agent is its multi-basic nature [64], which makes the coordination of metal cations possible [65]. These urea–cation complexes are the key to converting urea into a highly electrophilic “masked” isocyanate species [66,67], where the resonance is strongly reduced by the interaction with the metal center. This allowed the easy nucleophilic attack by Gly, as shown in Figure 1. The process is reversible [68] and proceeds stepwise, with the formation of a carbamic acid intermediate (**I**) (Figure 1) [69], which is converted in the final product through the elimination of a second ammonia molecule [70].

Urea is industrially prepared from ammonia and CO_2_ [71], therefore, it can be considered a CO_2_ carrier [72] and promises additional advantages regarding the overall carbon balance, especially when the proper recycling of ammonia is implemented. In other words, the upgrading of Gly to GC involves the fixation of one mole of CO_2_ per mole of vegetable oil (i.e., ~5 wt%, when 880 g/mol is taken as a mean molecular mass of a vegetable oil), a fact that, together with refined farming practices [3], can give additional support to reducing GHG emissions within the whole biodiesel industry.

In this paper, we present a novel continuous-flow process able to convert Gly into GC. The influence of key operating variables, such as time, temperature, and urea/Gly molar ratio (MR), on the product yield and selectivity are studied and statistically modeled by means of a Design of Experiments (DoE) approach.

## 2. Materials and Methods

### 2.1. General Information

Solvents and reagents were commercial grade and used as received. Glycerol (from vegetable source) was purchased from Merck (Milano, Italy). ZnSO_4_·H_2_O was obtained by heating ZnSO_4_·7H_2_O overnight in an oven at 130 °C, which was then stored in a desiccator. Gly was vacuum dried for 4 h in the rotavapor (water bath 40 °C) and stored in a capped bottle. Urea was dried at 65 °C overnight and stored in a desiccator. ^1^H NMR spectra were acquired with a Bruker Avance 400 spectrometer (Billerica, MA, USA).

### 2.2. Procedure for Preliminary Batch Reactions

In a 10 mL Schlenk tube with a screw cap and equipped with a stirring bar, Gly (1.0 g, 10.9 mmol), solid catalyst (0.05 mol/mol Gly, see later), and urea (0.82 g, 13.65 mmol, 1.25 mol/mol Gly) were inserted. A membrane vacuum pump was connected through the side arm and the vacuum was set at 400 mmHg. The capped tube was inserted in an oil bath (pre-heated at 150 °C) and stirred for 4 h. The crude reaction mixture, once cooled to room temperature (r.t.), was extracted with EtOAc:Et_2_O (4:1 mixture, 3 × 5 mL), and the crude colorless product, obtained from the dried organic phase, was analyzed with ^1^H NMR using CDCl_3_ or D_2_O as the solvent.

### 2.3. Procedure for Continuous Flow Reactions

The plant assembly is depicted in Figure 2. The reacting mixture was continuously recirculated through a heated tubular reactor (R) by means of a Bellco (model: BL 758, Mirandola, Italy) peristaltic pump (P). A mixing chamber (M), composed of a 25 mL two-necked round bottom flask, was heated and stirred through a standard stirring plate and an oil bath. A three-arm distillation connector acted as an expanding chamber (E) and connected the reactor output to an air-cooled condenser (A, also connected to M) and a vacuum line, set at 400 mm Hg. The said reactor was composed of a metallic AISI 316 stainless steel tube (1/16 in od × 1.2 mm id, 3 m long, ~4 mL internal volume, sourced from Restek, Milano, Italy) coiled around a 4 cm diameter cylindrical aluminum block, featuring a slot for a heating resistor and temperature sensor. Both of these heating and measuring elements were connected to a PID controller (Rex-C100, sourced from RobotDigg, Shangai, China), which allowed for setting and maintaining the desired temperature. Some thermal insulation (not shown in figure) was wrapped around the external side of the reactor.

Gly (11.0 g, 0.119 mol), ZnSO_4_·H_2_O (0.643 g; 3.58 mmol, 0.03 mol/mol Gly), and urea (e.g., 8.97 g, 0.149 mol for 1.25:1.00 urea:Gly molar ratio) were stirred at 65 °C within the mixing chamber (M, Figure 2) until a homogeneous solution was obtained. By that time, the temperature of the coiled reactor (R) was set at the desired value. Then, the peristaltic pump was started at constant flow (3.0 mL/min), and the mixture was recirculated for the desired time. To assess yield and conversion of each experiment, a 2.0 g portion of the crude reaction mixture was extracted with EtOAc:Et_2_O (4:1 mixture, 3 × 5 mL), and the raw colorless product, obtained from the organic phase, was carefully dried at the rotavapor. Purity was directly assessed with ^1^H NMR spectroscopy (see Appendix A). GC gave the following signals (Appendix A): ^1^H NMR (400 MHz, CDCl_3_, 298 K): *δ*_H_ (ppm) = 4.81 (1H, CH, m), 4.49 (2H, CH_2_, m), 3.99 (1H, CH_2_, dd, *J* = 12.84 Hz; 3.1 Hz), 3.72 (1H, CH_2_, dd, *J* = 12.84; 3.5 Hz), 2.21 (1H, OH, br). The metal catalyst and some carbamic acid intermediates (currently not characterized) constituted the denser, polar phase insoluble in the extracting mixture. No residual Gly was detected.

### 2.4. Design of Experiments (DoE)

Nineteen experiments were suggested, and the relative output data were statistically analyzed by means of Design Expert^®^ v.12 software (Stat-Ease Inc., 1300 Godward Street NE, Suite 6400, Minneapolis, MN 55413 USA). A three-level face-centered cubic design with five replicates of the central point was implemented. The investigated independent parameters were time (90–210 min), temperature (175–195 °C), and urea/Gly ratio (1.2–1.8), while output parameters were GC yield (%) and GC selectivity (%). Reagent flow was demonstrated to be not significant during the preliminary experiments, therefore, it was kept constant at 3.0 mL/min. Pressure was set to 400 mmHg because of hardware technical limits (squeezing of peristaltic pump tube) as well as urea losses through sublimation.

## 3. Results

Gly carbonylation with urea commonly involves solventless operation, temperatures ranging from 120 °C to 160 °C, reaction times from 3 to 24 h, and the presence of a catalyst (metal salt or metal oxide). Some degree of vacuum is also commonly applied in order to favor the removal of ammonia, obtaining a desired equilibrium shift [73,74] (Figure 1). A number of catalysts were described for this transformation [75,76,77,78] and, among these, zinc-containing species are by far the most active and popular. Irrespective of its initial form, soluble zinc-based catalysts are expected to give the same performance thanks to the formation of the metal glycerolate [79,80,81], while insoluble forms [67,82,83,84] are involved in heterogeneous reactions.

It is interesting to note that most of the recent papers have focused on an accurate description of the catalyst but have not given enough attention to the separation and isolation of GC, which represent key steps for considering the industrial feasibility of the chemical process [74].

Our investigations started with some preliminary experiments in which mixtures composed of Gly:urea:catalyst in a 1.00:1.25:0.05 molar ratio (MR) were heated in batch at 150 °C for 4 h, keeping the pressure at 400 mmHg (see materials and methods). After extraction with organic solvents, the amount of GC was evaluated with ^1^H NMR spectroscopy [85]. This screening, of which the results are collected in Table 1, let us draw some general observations, such as (i) yields are limited to 30%; (ii) longer reaction times do not improve the result (entry 3 vs. 2, Table 1); (iii) temperatures higher than 150 °C induce degradation (reaction mixture became brown and lower yields of GC were obtained); (iv) pressures lower than 400 mmHg induce urea losses through sublimation [86]; (v) catalyst anhydrification is beneficial (entry 2 vs. 1).

Cheap and easily available ZnSO_4_·H_2_O was chosen as the reference catalyst, especially because of its ability to give homogeneous mixtures that, being simpler reacting systems, let us focus on engineering approaches towards process improvement.

Some attempts to set up a reactive distillation [87] were carried out, however, the high boiling point of GC (~140 °C at 0.5 mmHg) was a considerable hurdle to its isolation. Better success comes from the known implementation of microwave-heated batch reactors, giving improved isolated yields and reduced reaction times [88,89]. As this suggests, a main problem of the batch process featuring traditional heating is likely to be the supply of thermal energy to the reacting mixture that, being subjected to an endothermic event [73], is affected by self-cooling, especially in the core (far from the heating bath). Therefore, to improve such heat transfer limitations, our focus was directed to the implementation of a high area-to-volume ratio reactor, such as a small diameter tubular reactor, together with a continuous flow of reactants inside it.

Solventless operation is a valuable feature of this process, but the high viscosity of Gly/urea mixtures poses serious troubles concerning their pumpability. Different HPLC pumps were not able to give a reliable flow, possibly due to the inability of check valves to promptly block the backflow. The addition of a high boiling point solvent such as DMSO was proposed as a solution [90], but we judge this practice undesirable as it nullifies the “solventless advantage” and makes the isolation of reaction products troublesome. Fortunately, the use of a peristaltic pump in addition to a heated mixing chamber (M, Figure 2) allowed us to obtain a steady flow of reactants. In fact, by keeping the Gly/urea mixture at 65 °C, a useful reduction in viscosity was observed and, more importantly, the precipitation of urea within the tubes was avoided.

Using the flow chemistry technique in small volume reactors, thanks to strongly increased heat and mass transfer, typically benefits enhanced reaction control with respect to batch processes, especially in fast reactions [91]. Nevertheless, the process under study (Figure 1) features slow kinetics as evidenced by the long reaction times required for complete conversion, such as 2 h at 150 °C with microwave irradiation [89]; the meagre conversion obtained with reaction times of a few minutes, even with the implementation of a capillary reactor [90], gives evidence of that. Using a coiled 3 m long tubular reactor, we first tried to get the best conversion through a single pass, keeping flow at a minimum (0.2 mL/min) and evaluating increased temperatures. Unluckily, unsatisfactory conversion was obtained even at 195 °C, while greater temperatures resulted in brown (degraded) mixtures, containing low amounts of GC. A possible solution was to consider multiple passes through the reactor, thus, a recirculating layout was set up, as shown in Figure 2.

Gly, urea, and the catalyst were stirred at 65 °C in the mixing chamber (M, Figure 2) to obtain a homogeneous mixture. This was continuously pumped through the heated coiled tubular reactor (R, Figure 2) and the outflow was recirculated through an air-cooled condenser (A) into the mixing chamber. The total volume of the reacting mixture was set to ~1.2 times of the total piping volume (reactor and interconnections) to maximize the number of passes through R. The entire system was maintained at reduced pressure by a membrane pump connected to E. At time intervals, a sample of reacting mixture was withdrawn from the mixing chamber to assess conversions and yields. Several trials let us delineate the general features of the system, as follows:temperatures greater than 175 °C resulted in better GC yields;a minimum time of 90 min was required to obtain complete conversion of Gly;pressures lower than 400 mmHg resulted in unreliable flow due to peristaltic pump malfunction (tube squeezing) and minor urea losses due to sublimation;yields were unaffected by changes in flow rate, within the range from 0.5–5.0 mL/min;yields were unaffected by changes in catalyst amount, within the range from 0.03–0.05 mol_ZnSO4·H2O_/mol_Gly_;diglycerol tricarbonate (DGTC) was identified as the main by-product [23].

In order to get a better understanding of the influence of multiple process parameters, we decided to implement a multivariate statistical evaluation based on a DoE approach. In particular, a three-level face-centered central composite design (CCD) was chosen and was used to define an appropriate number of experiments within the variable domains arising from the above observations. The included independent variables were the reactor temperature (ranging from 175 to 195 °C), the recirculation time (from 90 to 210 min), and the urea:Gly MR (from 1.2 to 1.8); while the other process parameters such as pressure, flow, amount of catalyst, and mixing chamber temperature were fixed at 400 mmHg, 3.0 mL/min, 0.03 mol_ZnSO4·H2O_/mol_Gly_, and 65 °C, respectively. As suggested by the DoE CCD model, we planned nineteen experiments, with five replicates of the central point, as shown in Table 2. All the experiments were conducted by the same operator to minimize systematic errors, while the order of experiments was randomized. The two monitored responses were GC yield and GC purity. A quantitative evaluation of the reaction selectivity (100·mol GC/(mol GC + mol DGTC)) was obtained by the integration of isolated ^1^H NMR signals of the product and isolated ^1^H NMR signals of the main by-product (DGTC), as described in the Appendix A. This was supported by the fact that the ^1^H NMR signals due to other substances always presented at a very low intensity.

## 4. Discussion

The ability to statistically evaluate the interrelation between process variables is one of the main advantages offered by the DoE approach. For instance, the influence of the three independent variables on the output responses are shown in Figure 3. GC-isolated yields (Figure 3a–c) strongly depends upon recirculation time and reactor temperature, while less marked is the influence of the urea:Gly MR. Best GC yields are obtained at temperatures between 180 and 190 °C, times between 110 and 160 min, and for urea:Gly MR in the range from 1.3–1.7. GC selectivity (Figure 3d–f) strongly depends upon urea:Gly MR and reactor temperature, while recirculation time is found to be the less influent parameter. Best selectivities are obtained at lower temperatures, lower urea:Gly MR, and times between 110 and 170 min.

The simultaneous influence of two independent variables on each physical property is of particular value. Figure 4 shows some “two factors response surfaces”, which put into evidence the dependency of GC yield% (left of Figure 4) from time–temperature or urea:Gly MR–time couples. On the right of the same figure, the influence of time–temperature or ratio–temperature couples on GC selectivity% are shown.

Finally, both GC yield% and GC selectivity% response parameters can be conveniently composed in a single factor, called “desirability”, which is able to describe the best process conditions in a simple and effective way. In the present case, desirability is obtained by the product of the said response parameters, normalized to unity. A desirability of 1.0 means that both parameters are maximized, while the lowest desirability (0.0) describes conditions where both responses are at minimum levels. In Figure 5, some contour heatmap plots of desirability as a function of independent variable couples are shown.

The strong reduction in process performance at high urea:Gly MR (Figure 5a,b) comes from a loss of selectivity (see also Figure 3b,e), meaning an increased formation of DGTC. This behavior is not surprising, as DGTC is described as an overreaction product of GC when the carbonylation reagent is in high molar excess [23,92,93]. This also suggests good potential for DGTC production as a possible future implementation of the process. The significant yield decrement at high times and temperatures (Figure 3 and Figure 5c) found good correlation with the experimental observation of the brownish color acquired by the reacting mixture in these circumstances, a sign of partial degradation. Overall, best operating conditions of the process were found to be a reactor temperature from 180 to 185 °C, a recirculation time from 120 to 150 min, and a urea:Gly MR of 1.25.

Regarding the potential risk of isocyanic acid emission coming from carbamate decomposition [94,95], the process conditions (low operating temperatures, low urea:Gly MR, presence of glycerol) make this event highly unlikely. As an additional safety measure, it might be considerable, for further developments, to add a “water trap” on the output of the vacuum pump.

## 5. Conclusions

The most challenging issue we faced during the development of the process was the pumping of the viscous reacting mixture. This was solved by means of a peristaltic pump (P, Figure 2) and by a heated mixing chamber (M) able to lower the viscosity and increase urea solubility. The technical limits of this implementation were evidenced when we attempted to operate at a pressure lower than 400 mmHg. This pressure constraint is thought to be one of the factors limiting the GC-isolated yield to ~42%; therefore, the employment of different pumping hardware could significatively improve performance. Moreover, to the best of our knowledge, this is the first example of a multiple pass tubular reactor applied to the conversion of Gly into GC. This plant layout (Figure 2) allowed for the management of a kinetically slow process within a confined, highly thermally controlled reactor (R). The same layout also features a closed loop with a single evacuation point (E), which has the particular advantage of collecting the co-produced ammonia and recycling it to urea (Figure 1). This feature, coupled with the modest molar excess of urea required for the transformation, demonstrate the opportunity to sustainably transfer an equimolar amount of CO_2_ to Gly, beneficially contributing to the entire carbon cycle of the biodiesel industry.

## Figures and Tables

**Figure 1 bioengineering-09-00778-f001:**
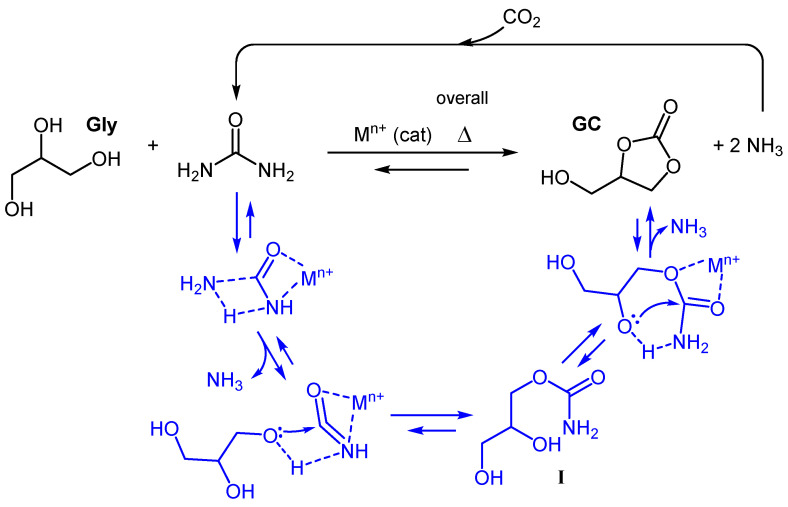
Metal-catalyzed carbonylation of Gly with urea.

**Figure 2 bioengineering-09-00778-f002:**
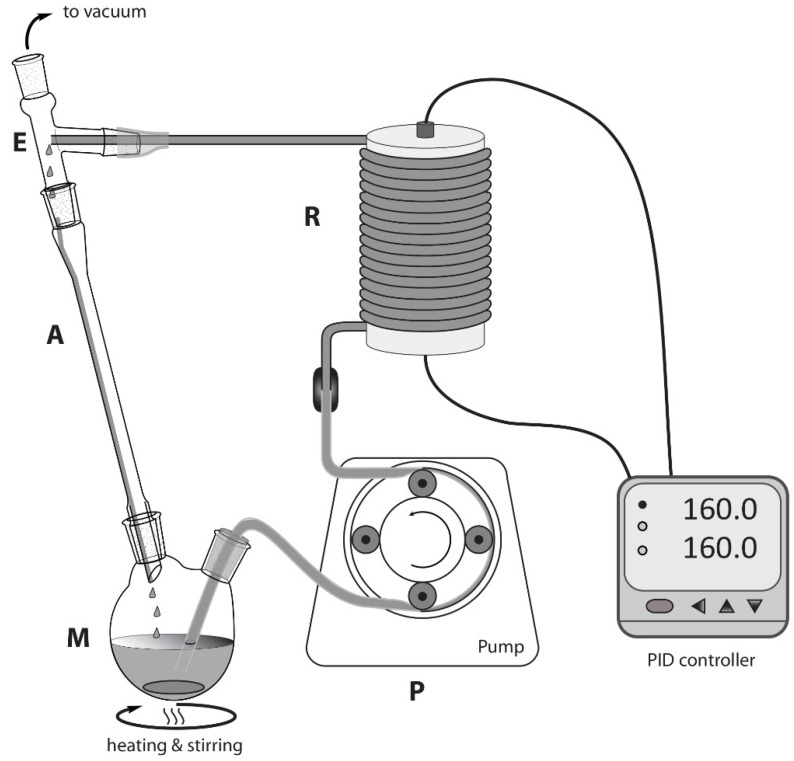
Layout of the continuous flow reactor. M = mixing chamber; P = peristaltic pump; R = coiled tubular reactor; E = expansion chamber; A = air-cooled condenser.

**Figure 3 bioengineering-09-00778-f003:**
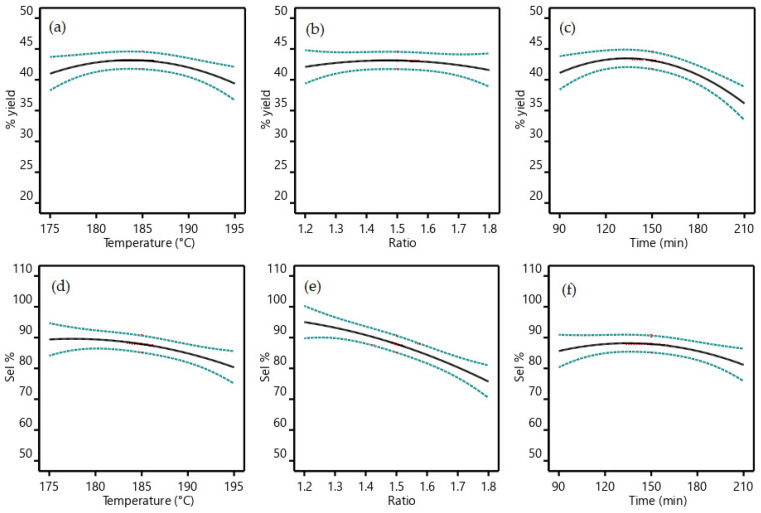
All factors graphs. (**a**) GC yield% vs. temperature, (**b**) GC yield% vs. urea/Gly MR, (**c**) GC yield% vs. recirculation time, (**d**) GC selectivity% vs. temperature, (**e**) GC selectivity% vs. urea/Gly ratio and (**f**) GC selectivity% vs. recirculation time. Dashed blue lines refers to minimum and maximum values of the data set. Continuous black lines refers to the mean values.

**Figure 4 bioengineering-09-00778-f004:**
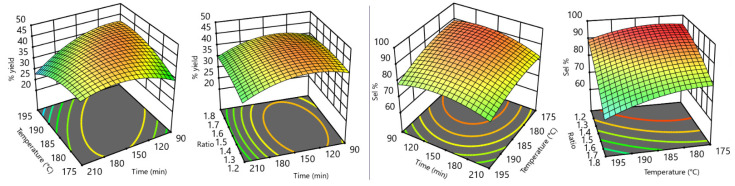
Two factors response surfaces.

**Figure 5 bioengineering-09-00778-f005:**
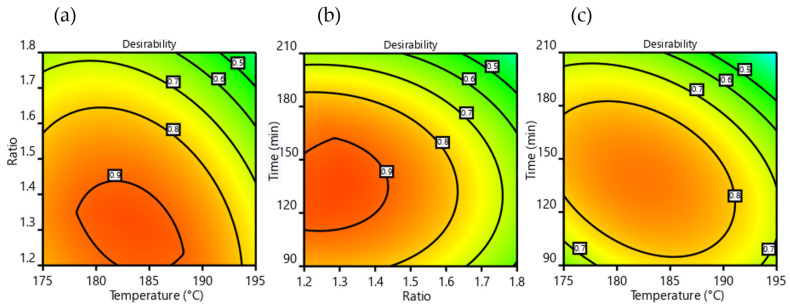
Desirability contour maps, (**a**) vs. urea/Gly MR-temperature, (**b**) vs. time-urea/Gly MR and (**c**) vs. time-temperature.

**Table 1 bioengineering-09-00778-t001:** Screening of alternative catalysts in the batch conversion of Gly to GC ^a^.

Entry	Catalyst	Isolated Yield (%)
1	ZnSO_4_·7H_2_O	27
2	ZnSO_4_·H_2_O	30
3 ^b^	ZnSO_4_·H_2_O	27
4	ZnCl_2_	30
5	FeCl_3_	20
6	MgO	22

^a^ Gly (1.00 g, 10.9 mmol), urea (0.815 g, 13.6 mmol), catalyst (5 mol% resp. to Gly), 150 °C, 4 h, 400 mmHg. ^b^ reaction time extended to 6 h.

**Table 2 bioengineering-09-00778-t002:** Experiments suggested by the DoE model ^a^.

Exp. n.	Temperature (°C)	Urea:Gly MR	Time (min)	GC Yield (%)	GC Selectivity (%)
1	195	1.8	210	25.8	57
2	175	1.2	90	33.6	93
3	175	1.8	210	37.9	68
4	185	1.5	210	35.2	85
5	185	1.2	150	40.3	91
6	175	1.2	210	37.9	90
7	195	1.2	90	41.9	86
8	185	1.5	150	42.9	88
9	185	1.8	150	42.3	80
10	175	1.8	90	33.4	82
11	195	1.2	210	27.0	84
12	185	1.5	150	45.1	88
13	185	1.5	150	45.7	86
14	185	1.5	150	42.8	89
15	185	1.5	150	41.3	89
16	195	1.8	90	38.7	64
17	175	1.5	150	39.0	87
18	185	1.5	90	41.0	82
19	195	1.5	150	40.3	83

^a.^ fixed parameters: P = 400 mmHg; flow = 3.0 mL/min; catalyst amount 0.03 mol_ZnSO4·H2O_/mol_Gly_.

## Data Availability

The data presented in this study are available in the article and Appendix A.

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
