# Peer review of "Upgrading of Biobased Glycerol to Glycerol Carbonate as a Tool to Reduce the CO2 Emissions of the Biodiesel Fuel Life Cycle"

_bioengineering, 2022, doi:10.3390/bioengineering9120778_

Round 1

Reviewer 1 Report

The manuscript “Upgrading of Biobased Glycerol to Glycerol Carbonate as a tool to reduce the CO2 emissions of the biodiesel fuel life cycle” deals with the investigation of the production of glycerol carbonate by urea alcoholysis under an optimised continuous-flow, lab-scale, reactor. Several reaction parameters were taken into account and optimised with a DoE approach. The reaction is of course of great interest, potentially able to valorise at the same time both glycerol and, indirectly, CO2; moreover the manuscript is well written and the results professionally discussed. Therefore, this manuscript deserves to be considered for publication in Bioengineering after few, but fundamental improvements and clarifications as described below.

1.  is ZnSO4 completely soluble in glycerol and urea mixture at 65°C? Could ZnSO4 be recovered after the reaction or after the reaction Zn is in a different form (i.e. glycerolate) thus impeding its recyclability? Have the authors tried some tests by recycling the recovered Zn-contained solid/solution?

2.    In line 211 authors states that “ii) longer reaction times do not improve the results”; however in table 1 there are no tests which support this statement.

3.    Linked to the previous question: why the reagent flow is not significant for the outcome? It will change the residence time inside the heated part of the flow reactor therefore changing the reaction time. What is the ratio between the volume of the heated part of the reactor (part “R” in figure 2) and the total volume of the system? What is the volume of the solution initially loaded in “M” (Figure 2), how many cycles it performs under the best conditions reported in table 2? Authors should comment on that in the main text.

4.    Finally, the main safety issue of the alcoholysis of urea for the production of organic carbonates is the potential unselective degradation of the carbamate intermediate to isocyanic acid (HNCO): a colourless, volatile and poisonous substance, with a boiling point of 23°C. Can the author exclude the formation of this by-product and its continuous removal through the membrane vacuum pump?

Reviewer 2 Report

Review of paper ‘Upgrading of Biobased Glycerol to Glycerol Carbonate as a tool to reduce the CO2 emissions of the biodiesel fuel life cycle’ prepared by Biagio Anderlini, Alberto Ughetti, Emma Cristoni, Luca Forti, Luca Rigamonti, and Fabrizio Roncaglia.

The paper bioengineering-2050685 presents work related to the important aspect of reducing greenhouse gas emissions. To this end, the authors have proposed the use of waste glycerol. I have some suggestions that authors may consider before publishing this work.

1. English needs to be corrected. The entire manuscript should be revised. Here are some proposed changes to the Introduction:

In addition to different natural dynamics outside of our control like solar activity, the development of the last hundred years has been accompanied by a general and progressive increase of the mean global temperature, collectively known as ‘global warming’. This variation is strictly related to several climate alterations such as melting glaciers and rising sea levels, but also with extreme weather events, a change in wildlife habitats, and an array of other impacts [1]. Beyond the heating due to direct irradiation from the Sun, the atmosphere plays a determinant role in retaining part of the thermal energy close to the Earth’s surface.

 2. In the Materials and methods section, please indicate clearly the source of the glycerol derivation. Is it from biodisel production or purchased as a reagent? This is an important aspect, as it determines the practical application of the solution proposed by the authors. At the same time, if it is biobased glycerol, please specify its characteristics and origin, if not the title should be corrected.

3. The authors pointed out the optimum working conditions but did not address an important issue, from a practical point of view. It is very often the case that optimum conditions in terms of process efficiency do not represent the best conditions in terms of, for example, financial outlay or design solutions. For example, the authors point out a problem with the viscosity of the solutions, but with a change in temperature the viscosity will change. On the other hand, a change in temperature (both cooling and heating) causes a significant increase in operating costs. I recommend listing the factors that may be relevant when implementing this solution in practice.
4. The authors should relate their results to those described in the literature and patents on carbon dioxide removal methods. In this way, it will be possible to identify the added value of the proposed solution.
